# BLOCK-LOCAL LEARNING WITH PROBABILISTIC LATENT REPRESENTATIONS

## ABSTRACT

The ubiquitous backpropagation algorithm requires sequential updates through the network introducing a locking problem. In addition, backpropagation relies on the transpose of forward weight matrices to compute updates, introducing a weight transport problem across the network. Locking and weight transport are problems because they prevent efficient parallelization and horizontal scaling of the training process. We propose a new method to address both these problems and scale up the training of large models. Our method works by dividing a deep neural network into blocks and introduces a feedback network that propagates the information from the targets backwards to provide auxiliary local losses. Forward and backward propagation can operate in parallel and with different sets of weights, addressing the problems of locking and weight transport. Our approach derives from a statistical interpretation of training that treats output activations of network blocks as parameters of probability distributions. The resulting learning framework uses these parameters to evaluate the agreement between forward and backward information. Error backpropagation is then performed locally within each block, leading to "block-local" learning. Several previously proposed alternatives to error backpropagation emerge as special cases of our model. We present results on a variety of tasks and architectures, demonstrating state-of-the-art performance using block-local learning. These results provide a new principled framework for training networks in a distributed setting.

## 1 INTRODUCTION

Recent developments in machine learning have seen deep neural network architectures scale to billions of parameters (Touvron et al., 2023; Brown et al., 2020). While this has increased the power of these models to unprecedented levels, it has also pushed the computing hardware on which large network models run to its limits. As a result, it has become increasingly important to distribute the model training across many independent computing devices. However, today's machine learning algorithms are poorly suited for distributed training. The error backpropagation algorithm requires an alternation of interdependent forward and backward phases, each requiring sequential computation. This introduces a locking problem because each phase must wait for the other (Jaderberg et al., 2016). Furthermore, the two phases rely on the same weight matrices to compute updates, which makes it impossible to separate memory spaces. This is referred to as the weight transport problem, see Grossberg (1987); Lillicrap et al. (2014a). Locking and weight transport are problems because they make efficient parallelization and horizontal scaling of large machine learning models across compute nodes extremely difficult.

We propose a new method to address these problems that distributes a globally defined optimization algorithm across a large network of computing devices using only local updates for training. Our approach utilizes a variational inference approach that uses results from probabilistic models to provide auxiliary local targets from a separate feedback network that propagates information from the targets to the input. Thus, messages can be communicated forward and backwards between computational nodes in parallel and include information about extracted features, which are updated using local probabilistic losses calculated using the targets provided by the feedback network. In contrast to previous results, optimizing these local losses does not require a contrastive step where different positive and negative samples are propagated through the network. Within each block, conventional error backpropagation is performed locally ("block local") both in the forward network

and the backward feedback to adapt parameters during training. Performing forward and backward propagation in parallel mitigates the locking problem, and having a separate feedback network solves the weight transport problem.

The learning model developed here provides a new principled method for distributing the training of networks across multiple computing devices. The solutions emerging from this framework show striking similarities to those of previous models that used random feedback weights to provide local targets (Lee et al., 2015; Meulemans et al., 2020; Lillicrap et al., 2020; Ernoult et al., 2022), but we provide a principled way to train these feedback weights.

In summary, the contribution of this paper is threefold:

1. We provide a theoretical framework for interpreting the representations of deep neural networks as parameters of probability distributions.

2. Based on this probabilistic framework, we derive a new variational bound that allows us to decompose the global log-likelihood loss into a sum of local terms, which provides a principled approach to block-local training of these networks.

3. We show that this probabilistic learning method can achieve state-of-the-art performance on several benchmark classification tasks.

## 2 RELATED WORK

A number of methods for using local learning in DNNs have been introduced previously. Random feedback alignment (Lillicrap et al., 2016) and related approaches (Akrout et al., 2019; Nøkland, 2016; Samadi et al., 2017) use fixed random feedback weights to back-propagate errors. Jaderberg et al. (2017) used layer-wise learned predictors of gradients, called "synthetic gradients" to decouple the training of different layers. Target propagation (Lee et al., 2015; Meulemans et al., 2020) has been demonstrated to have competitive performance using random projections for target labels instead of errors (Frenkel et al., 2021; Ernoult et al., 2022; Shibuya et al., 2023). Target Projection Stochastic Gradient Descent (tpSGD) Lomnitz et al. (2022) uses layer-wise SGD and local targets generated via random projections of the labels but does not adapt the backward weights. The network architecture used in our approach is similar to these prior works, but, in addition, provides a principled way to adapt feedback weights.

Some previous methods are based on probabilistic or energy-based cost functions. Jimenez Rezende et al. (2016) used a generative model and a KL-loss for local unsupervised learning of 3D structures. Contrastive learning (Chen et al., 2020; Oord et al., 2019) has been used to construct block-local losses (Xiong et al., 2020; Illing et al., 2021). Equilibrium propagation replaces target clamping with a target nudging phase (Scellier and Bengio, 2017). Another interesting contrastive approach, forward propagation, was recently introduced (Hinton, 2022; Zhao et al., 2023) which needs task-specific negative input examples. In contrast to these methods, our approach does not need separate positive and negative data samples and focuses on block-local learning. A number of methods have been proposed based on predictive coding framework (Millidge et al., 2022; Salvatori et al., 2022) but with a focus on biologically motivated generative models (Ororbia and Mali, 2019).

Other methods (Belilovsky et al., 2019; Löwe et al., 2019) have used greedy local, block- or layer-wise optimization. Notably, (Nøkland and Eidnes, 2019) achieved good results by combining a matching and a local cross-entropy loss. In contrast to our method, they used a similarity matching loss across mini-batches which prevents parallelization across a batch of data samples. Siddiqui et al. (2023) recently used block-local learning based on a contrastive cross-correlation metric over feature embeddings (Zbontar et al., 2021), demonstrating promising performance. Wu et al. (2021) used greedy layer-wise optimization of hierarchical autoencoders for video prediction. Wu et al. (2022) used an encoder-decoder stage for pre-training. In contrast to these methods, we do not rely solely on local greedy optimization but provide a principled way to combine local losses with feedback information without locking and weight transport across blocks and without contrastive learning.

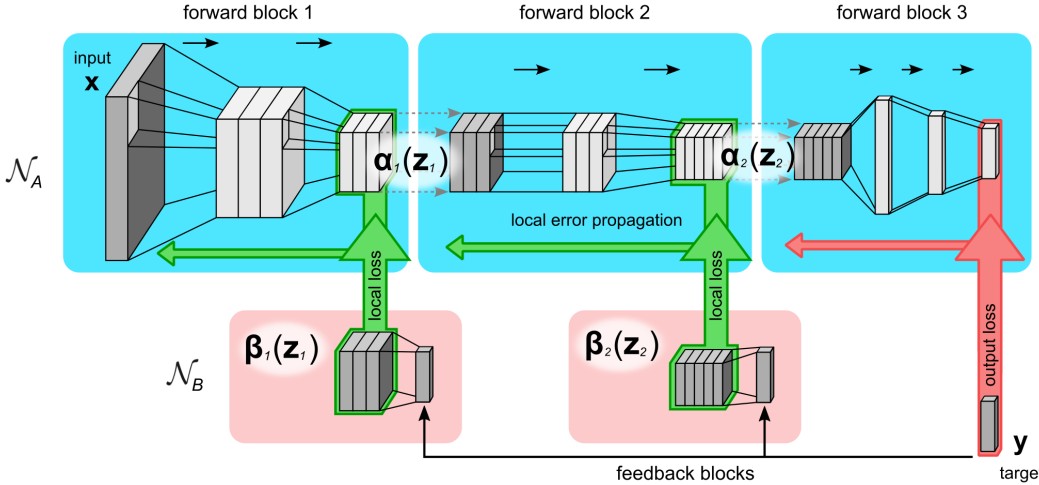

Figure 1: Illustration of use of block-local representations as learning signals on intermediate network layers. A deep neural network architecture $\mathcal{N}_A$ is split into multiple blocks (forward blocks) and trained on an auxiliary local loss. Targets for local losses are provided by a feedback backward network $\mathcal{N}_B$.

## 3 A PROBABILISTIC FORMULATION OF DISTRIBUTED LEARNING

In this section we establish a method to partition a deep neural network into blocks by interpreting activations as parameters of probability distributions. We use these intermediate probabilistic representations at each block to derive block-local losses. To do this, we introduce a feedback network that accompanies the feedforward network to compute probabilistic representations. We show that the derived block-local losses and the resulting block-local learning (BLL) can be realized by a posterior bootstrapping mechanism that combines forward and feedback activations.

### 3.1 USING LATENT REPRESENTATIONS TO CONSTRUCT PROBABILISTIC BLOCK-LOCAL LOSSES

Learning in deep neural networks can be formulated probabilistically (Ghahramani, 2015) in terms of maximum likelihood, i.e. the problem is to minimize the negative log-likelihood $\mathcal{L} = -\log p(\mathbf{x}, \mathbf{y}) = -\log p(\mathbf{y} \,|\, \mathbf{x}) - \log p(\mathbf{x})$ with respect to the network parameters $\boldsymbol{\theta}$. For many practical cases where we may not be interested in the prior distribution of the input $p(\mathbf{x})$, we would like to directly minimize $\mathcal{L} = -\log p(\mathbf{y} \,|\, \mathbf{x})$.

This probabilistic interpretation of deep learning can be used to define block-local losses and distribute learning across multiple blocks of networks by introducing intermediate latent representations. The idea is illustrated in Fig. 1. A neural network $\mathcal{N}_A$ computing the mapping $\mathbf{x} \rightarrow \mathbf{y}$ takes $\mathbf{x}$ as input and its outputs can be interpreted as the statistical parameters of the conditional distribution $p(\mathbf{y} \,|\, \mathbf{x})$. When the network is split at intermediate layers into blocks, training using end-to-end gradient estimation can be replaced by estimators that optimize the blocks $\mathbf{x} \rightarrow \mathbf{z}_1$, $\mathbf{z}_1 \rightarrow \mathbf{z}_2$ $\ldots \mathbf{z}_N \rightarrow \mathbf{y}$ separately. To see this, consider the gradient of the log-likelihood loss function

$$-\nabla\mathcal{L} = \nabla \log p(\mathbf{y} \,|\, \mathbf{x}) \ , \tag{1}$$

where $\nabla$ is the vector differential operator over parameters $\boldsymbol{\theta}$. For any deep network, it is possible to choose an intermediate activation at an arbitrary layer to represent a latent variable $\mathbf{z}_k$ so that $p(\mathbf{y} \,|\, \mathbf{x}) = \mathbb{E}_{p(\mathbf{z}_k \,|\, \mathbf{x}, \mathbf{y})}[p(\mathbf{y} \,|\, \mathbf{z}_k) \, p(\mathbf{z}_k \,|\, \mathbf{x})]$, where $\mathbb{E}_p[]$ denotes expectation with respect to $p$. Therefore, the representations of $\mathbf{y}$ depend on $\mathbf{x}$ only through $\mathbf{z}_k$, as expected for a feedforward network. Using this conditional independence property, the log-likelihood (1) expands to

$$-\nabla\mathcal{L} = \mathbb{E}_{p(\mathbf{z}_1 \ldots \mathbf{z}_N \,|\, \mathbf{x}, \mathbf{y})}\left[\nabla \log p(\mathbf{z}_1 \,|\, \mathbf{x}) + \nabla \log p(\mathbf{z}_2 \,|\, \mathbf{z}_1) + \cdots + \nabla \log p(\mathbf{y} \,|\, \mathbf{z}_N)\right] \ . \tag{2}$$

The identity in (2) is well known and also exploited in the derivation of the Expectation-Maximization (EM) algorithm (Dempster et al., 1977) (see Sec. S1.3 in the Supplement for a recap).

Computing the expectation with respect to $p\left(\mathbf{z}_k \mid \mathbf{x}, \mathbf{y}\right)$ corresponds to the E-step and calculating the gradients corresponds to the M-step. The sum inside the expectation separates the gradient estimators into parts: $\mathbf{x} \rightarrow \mathbf{z}_1 \ldots \mathbf{z}_N \rightarrow \mathbf{y}$. Importantly, the parts can have separate parameter spaces $\boldsymbol{\theta}_k^{(1)}$, $\boldsymbol{\theta}_k^{(2)}, \ldots, \boldsymbol{\theta}_k^{(N)}$ so that the gradient estimators become independent. This provides the core idea for how to split the training problem into smaller, and potentially more local sub-problems.

However, the E-step is impractical to compute for most interesting applications because of the combinatorial explosion in the state space of $\mathbf{z}_k$, which renders the expectation in Eq. (2) intractable. To get around this, we use a variational upper bound $\mathcal{F} \geq \mathcal{L}$ (Jordan et al., 1999). We introduce a feedback network with independent parameters (see Fig. 1), that is used to construct an auxiliary distribution $q\left(\mathbf{z}_k \mid \mathbf{x}, \mathbf{y}\right)$ to substitute the intractable posterior $p\left(\mathbf{z}_k \mid \mathbf{x}, \mathbf{y}\right)$. The variational loss $\mathcal{F}$ is then used to jointly minimize $\mathcal{L}$ together with the distance between $p$ and $q$. We demonstrate that this approach can be used to split gradients in a similar fashion to Eq. (2), yielding a distributed approximate solution to Eq. (1). In the next section, we describe how we construct the variational distribution $q$.

## 3.2 AUXILIARY LATENT REPRESENTATIONS

The probabilistic interpretation of hidden layer activity outlined above is valid under relatively mild assumptions, which we will establish here. It is important to note that at no point does the network produce samples of the implicit random variables $\mathbf{z}_k$; they are introduced here only to conceptualize the mathematical framework. Instead, at block $k$, the network outputs the parameters of a probability distribution $\alpha_k(\mathbf{z}_k)$ (e.g., means and variances if $\alpha_k$ is Gaussian). $\alpha_k(\mathbf{z}_k) = p\left(\mathbf{z}_k \mid \mathbf{x}\right)$ is the distribution over $\mathbf{z}_k$ for given inputs $\mathbf{x}$. The network thus translates $\alpha_{k-1} \rightarrow \alpha_k \rightarrow \ldots$ by outputting the statistical parameters of the conditional distribution $\alpha_k(\mathbf{z}_k)$ and taking the $\alpha_{k-1}(\mathbf{z}_{k-1})$ parameters as input. More specifically, the network implicitly calculates a marginal distribution

$$\alpha_k\left(\mathbf{z}_k\right) = p\left(\mathbf{z}_k \mid \mathbf{x}\right) = \mathbb{E}_{p\left(\mathbf{z}_{k-1} \mid \mathbf{x}\right)}\left[p_k\left(\mathbf{z}_k \mid \mathbf{z}_{k-1}\right)\right] = \mathbb{E}_{\alpha_{k-1}\left(\mathbf{z}_{k-1}\right)}\left[p_k\left(\mathbf{z}_k \mid \mathbf{z}_{k-1}\right)\right], \quad (3)$$

where $\mathbb{E}_p\left[\,\right]$ denotes expectation with respect to the probability distribution $p$. Consequently, the network realizes a conditional probability distribution $p\left(\mathbf{y} \mid \mathbf{x}\right)$ (where $\mathbf{x}$ and $\mathbf{y}$ are network inputs and outputs, respectively). Eq. (3) is an instance of the belief propagation algorithm to efficiently compute conditional probability distributions. If all blocks have a rich enough expressive power (e.g. sufficient number of hidden layers) an accurate representation of the mappings between distributions can be learned in the network weights through error back-propagation. Thus, the distributions $p\left(\mathbf{z}_k \mid \mathbf{x}\right)$ in the variational learning framework outlined above are realized simply by propagating inputs $\mathbf{x}$ through the forward network $\mathcal{N}_A$.

To construct the variational distribution $q$, we introduce the backward network $\mathcal{N}_B$, which propagates messages $\beta_k$ backward. Inferences about the posterior distribution $p\left(\mathbf{z}_k \mid \mathbf{x}, \mathbf{y}\right)$ for any latent variable $\mathbf{z}_k$ can be made using the belief propagation algorithm, which propagates messages $\alpha_k\left(\mathbf{z}_k\right)$ forward through the network using Eq. (3) and messages $\beta_k\left(\mathbf{z}_k\right) = q\left(\mathbf{y} \mid \mathbf{z}_k\right)$ backwards from the labels. In Section 4.2 we also experimented with a variant where feedback messages are propagated backward through a multi-layer network. In both cases the variational posterior can be computed up to normalization

$$\rho_k\left(\mathbf{z}_k\right) = q\left(\mathbf{z}_k \mid \mathbf{x}, \mathbf{y}\right) \quad \propto \quad p\left(\mathbf{z}_k \mid \mathbf{x}\right) q\left(\mathbf{y} \mid \mathbf{z}_k\right) = \alpha_k\left(\mathbf{z}_k\right) \beta_k\left(\mathbf{z}_k\right). \quad (4)$$

We make use of the fact that, through Eq. (3), the parameters of a probability distribution $p\left(\mathbf{z}_k \mid \mathbf{x}\right)$ are a function of the parameters of $p\left(\mathbf{z}_i \mid \mathbf{x}\right)$, for $0 < i < k$, e.g. if $\alpha$ is assumed to be Gaussian we have $\left(\mu\left(\alpha_k\right), \sigma^2\left(\alpha_k\right)\right) = f\left(\mu\left(\alpha_i\right), \sigma^2\left(\alpha_i\right)\right)$, where $\mu\left(.\right)$ and $\sigma^2\left(.\right)$ are the mean and variance of the distribution respectively. Thus, if a network outputs $\left(\mu\left(\alpha_i\right), \sigma^2\left(\alpha_i\right)\right)$ on layer $i$ and $\left(\mu\left(\alpha_k\right), \sigma^2\left(\alpha_k\right)\right)$ on layer $k$, a suitable probabilistic loss function will allow the network to learn $f$ from examples. Therefore, the conditional distributions $p_k\left(\mathbf{z}_k \mid \mathbf{z}_{k-1}\right)$ and the expectation in Eq. (3) are only implicitly encoded in the network weights. Clearly, the sub-networks that compute the transition from one latent variable to the next can have separated parameter spaces. We will use the exponential family of probability distributions for which this observation can be formalized more thoroughly, as described next.

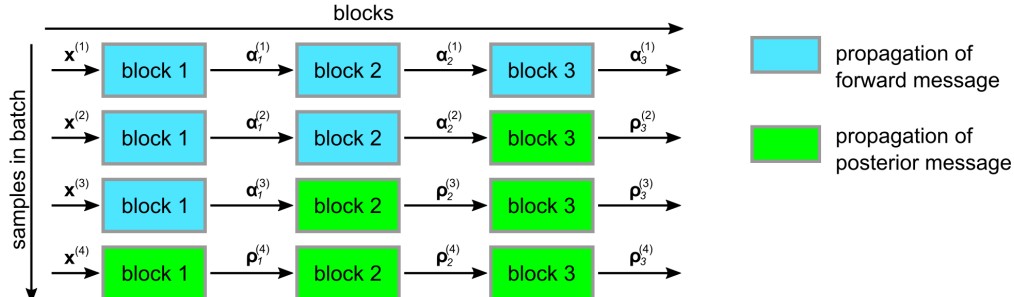

Figure 2: Illustration of posterior bootstrapping. Either the forward message $\alpha_k$ or the posterior message $\rho_k$ is propagated for each sample and block.

### 3.3 Exponential Family Distributions

To derive concrete losses and update rules for the forward and backward networks, we assume that $\alpha_k$'s and $\beta_k$'s are from the exponential family (EF) of probability distributions, given by

$$\alpha_k\left(\mathbf{z}_k\right) \;=\; \prod_j \alpha_{kj}\left(z_{kj}\right) \;=\; \prod_j h(z_{kj}) \exp\left(T\left(z_{kj}\right)\phi_{kj} - A\left(\phi_{kj}\right)\right) , \tag{5}$$

with base measure $h$, sufficient statistics $T$, log-partition function $A$, and natural parameters $\phi_{kj}$. This rich class contains the most common distributions, such as Gaussian, Poisson or Bernoulli, as special cases. For the example of a Bernoulli random variable we have $z_{kj} \in \{0, 1\}$, $T\left(z_{kj}\right) = z_{kj}$ and $A\left(\phi_{kj}\right) = \log\left(1 + e^{\phi_{kj}}\right)$ (Koller and Friedman, 2009). One interesting property of the EF is that the Kullback-Leibler (KL-) divergence, to measure the distance between two distributions $\rho_k$ and $\alpha_k$, with parameters $\gamma_k$ and $\phi_k$, can be expressed using only the means $\mu$ and variances $\sigma^2$ of the distributions, i.e.

$$-\nabla \mathcal{D}_{KL}\left(\rho_k \,|\, \alpha_k\right) \;=\; \sum_j \left(\mu\left(\rho_{kj}\right) - \mu\left(\alpha_{kj}\right)\right)\nabla\phi_{kj} \;+\; \sigma^2\left(\rho_{kj}\right)\left(\phi_{kj} - \gamma_{kj}\right)\nabla\gamma_{kj} . \tag{6}$$

We will exploit this property to construct local learning rules that can be computed efficiently. A network directly implements an EF distribution if the activations $a_{kj}$ at block $k$ encode the natural parameters, $a_{kj} = \phi_{kj}$.

To summarize, a feed-forward DNN $\mathcal{N}_A : \mathbf{x} \to \mathbf{y}$, can be split into $N + 1$ blocks by introducing implicit latent variables $\mathbf{z}_k : \mathbf{x} \to \mathbf{z}_k \to \mathbf{y}$, and generating the respective natural parameters. Blocks can be separated after any arbitrary layer, but some splits may turn out more natural for a particular network architecture. If both distributions $\alpha_{kj}$ and $\beta_{kj}$ are (assumed to be) members of the EF with natural parameters $\phi_{kj,\alpha}$ and $\phi_{kj,\beta}$, then $\rho_{kj}$ is also EF with parameters $\frac{1}{2}\left(\phi_{kj,\alpha} + \phi_{kj,\beta}\right)$ [1].

### 3.4 Modularized Learning Using Local Variational Losses And Posterior Bootstrapping

We construct and use an upper bound on the actual log-likelihood loss $\mathcal{L}$ for training the model. This upper bound consists only of block-local losses $\ell$ at all network blocks and is constructed using the forward and feedback networks $\mathcal{N}_A$ and $\mathcal{N}_B$, respectively, as shown in the Supplement. The local loss $\ell$ can be written as

$$\ell\left(p_k, \beta_k \,|\, \alpha_{k-1}\right) \;=\; \mathcal{D}_{KL}\left(q_k \,|\, \alpha_k\right) \;+\; \mathcal{H}\left(p_k \,|\, \alpha_{k-1}\right) , \tag{7}$$

where the first divergence term measures the mismatch between the posterior $\rho_k$ and the forward message $\alpha_k$, and the second term is an entropy loss that determines the quality of the distributions when propagating data through $\mathcal{N}_A$, based on the variational posterior (see Supplementary Information S1.2.1 for details).

---

[1]The dependence is linear and can be augmented with an arbitrary constant factor. $\frac{1}{2}$ is conveniently chosen here because it assures that forward messages and posterior parameters are of the same scale.

The loss in Eq. (7) is local in the sense that it is completely determined by the information available at block $k$, i.e., the local network transfer function specifying $p_k$, the forward message from the previous block $\alpha_{k-1}$, and the feedback $\beta_k$. Furthermore, the loss is local with respect to learning, i.e. it doesn't require global signals to be communicated to each block. In this sense, our approach differs from previous contrastive methods that need to distinguish between positive and negative samples. In our approach, any sample that passes through a block can be used directly for weight updating and is treated in the same way.

To arrive at this key result we use a new approach that we call "posterior bootstrapping". Posterior bootstrapping combines the information provided by the forward and backward network during learning by propagating one of either the forward message $\alpha_k$ or the parameters to the posterior message $\rho_k$ to the next block. Whether $\alpha_k$ or $\rho_k$ is passed for every sample and every block is determined by a bootstrapping schedule. The optimal schedule is derived in Supplementary Information S1.2.1 and is shown in Fig. 2, where the pattern of posterior propagation forms a block-triangular matrix, giving blocks close to the input a tendency to preferentially propagate forward messages. Computing the posterior in this EF model is computationally very cheap as outlined above, so it introduces no significant overhead. Posterior bootstrapping also does not introduce a locking problem because the backward messages $\mathbf{b}_k$ are simultaneously available at all blocks.

Based on posterior bootstrapping and optimization of $\ell$, it is possible to construct an unbiased learning algorithm for the networks $\mathcal{N}_A$ and $\mathcal{N}_B$. In Supplementary Information S1.2.1 we show the following theorem in detail

**Theorem 1.** *Let $\ell$ be the local loss function Eq.(7). Furthermore, let $\alpha_k^{(m)}$, $\beta_k^{(m)}$ be messages created using the posterior bootstrapping schedule outlined above. Then simultaneous minimization of $\ell\left(p_k^{(m)}, \beta_k^{(m)} \,\middle|\, \alpha_{k-1}^{(m)}\right)$ in all blocks $k$, minimizes an upper bound on the log-likelihood loss $\mathcal{L}$.*

The proof of Theorem 1 and additional details are presented in 2 steps in the Supplement. First we show that an upper bound to $\mathcal{L}$ can be constructed by adding loss terms of the form (7). We then show that posterior bootstrapping computes the expectations that are needed to provide the forward and backward messages $\alpha_k^{(m)}$ and $\beta_k^{(m)}$. In simulations we also tested a simpler bootstrapping schedule that just passes forward message $\alpha_k$ through the network.

### 3.5 Greedy local forward and feedback network optimization

We do the overall training of the model using a greedy block-local learning strategy, meaning that we treat the inputs as constants and do not apply the chain rule across block boundaries. We apply a greedy learning strategy to train the feedback network $\mathcal{N}_B$ as well. The role of the feedback network is to propagate information about the labels back to the blocks of the forward network, providing local targets for the losses $\ell$. The construction of the feedback network is therefore arbitrary and need not reflect the complexity of the forward network. In this paper, we use the simplest version, where each block of $\mathcal{N}_B$ is given by a single linear layer. This special case is of particular interest because it allows us use a closed form solution to the optimization problem to find the parameters of the feedback network that minimize $\ell$. In the Supplement Sec. S1.3 and S1.5 we show the closed-form solution is $\boldsymbol{\theta}_k^{(b)} \stackrel{!}{=} \arg\min_{\boldsymbol{\theta}_k^{(b)}} \ell\left(p_k, \beta_k \,|\, \alpha_{k-1}\right)$ and convergence properties.

### 3.6 Distributed variational learning

In summary, the BLL algorithm is given by Table 1. The two for loops can be interleaved and parallelized by pipelining the propagation of data samples through the network as shown in Fig. 3. Updates can be computed as soon as propagation through a given block is complete. There is no locking, since only the data labels are needed to compute the output of the backward network. Furthermore, there is no weight transport problem since parameter spaces are separated and updates are computed only locally.

BLL shares many similarities with earlier methods. In particular, Direct Feedback Alignment (DFA) propagates targets through random weights to create local learning targets and can therefore be seen as a special case of BLL where feedback weights are kept fixed and the number of blocks equals the number of layers in the model. The loss term that emerges in BLL also shows some similarity

**for** all pairs $\mathbf{x}, \mathbf{y}$ in the training data set, and learning rate $\eta$ **do**
    $\mathbf{a}_0 \leftarrow \mathbf{x}$
    **for** $1 \leq k \leq N$ **do**
        $\beta_k \leftarrow g_k(\mathbf{y})$                               $\triangleright$ *Feedback network*
        $\alpha_k \leftarrow f_k(\alpha_{k-1})$          $\triangleright$ *Forward network, computation depends on previous block*
        $\theta_k^{(b)} \leftarrow \arg\min_{\theta_k^{(b)}} \ell\left(p_k, \beta_k \,|\, \alpha_{k-1}\right)$
        $\theta_k^{(a)} \leftarrow \theta_k^{(a)} + \eta \, \nabla_{\theta_k^{(a)}} \, \ell\left(p_k, \beta_k \,|\, \alpha_{k-1}\right)$
        **if** (*posterior bootstrapping*) **then**
            $\alpha_k \leftarrow \rho_k$

Table 1: Pseudo code of the BLL training algorithm. The for loops can be interleaved and run in parallel. Colors correspond to the operations in Figure 3

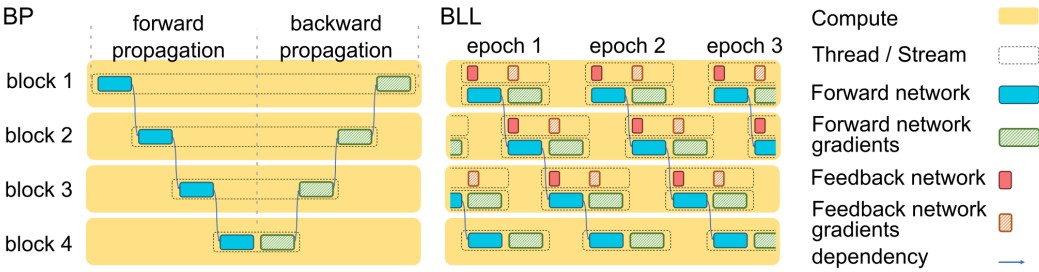

Figure 3: Timeline of execution for error backpropagation (BP) and BLL. BLL presented as a simplest case with forward-only bootstrapping.

with the predictive loss proposed in Nøkland and Eidnes (2019), but losses are derived here from a probabilistic framework and used to simultaneously learn the forward network and local targets.

# 4 RESULTS

## 4.1 BLOCK-LOCAL LEARNING OF VISION BENCHMARK TASKS

We evaluated the BLL algorithm on three vision tasks: Fashion-Mnist, CIFAR-10 and Imagenet-1K. Its performance is compared on ResNet18 and ResNet50 architectures with that of Backpropagation (BP), Feedback Alignment (Lillicrap et al., 2014b) (FA) and Local learning using similarity matching loss (Pred-Sim) from (Nøkland and Eidnes, 2019). The ResNet architectures were divided into

| Architecture | Algorithm | Fashion-MNIST | | CIFAR-10 | | ImageNet-1K | |
|---|---|---|---|---|---|---|---|
| | | test-1 | test-3 | test-1 | test-3 | test-1 | test-5 |
| ResNet-18 | BLL | 94.2 | 99.3 | 88.3 | 98 | | |
| | BP | 92.7 | 99.3 | 95.2 | 99.3 | | |
| | FA | 87.9 | 98.6 | 70.4 | 92.5 | | |
| | Pred-Sim | 93.9 | 99.3 | 88 | 97.7 | | |
| ResNet-50 | BLL | 94.3 | 99.1 | 92.6 | 99.1 | 53.6 | 77.1 |
| | BP | 93.4 | 99.4 | 94 | 99.2 | 76.1 | 92.9 |
| | FA | 83.1 | 97.9 | 70.3 | 92 | | |
| | Pred-Sim | 94.3 | 99.6 | 92.4 | 98.8 | | |

Table 2: Classification accuracy (% correct) on vision tasks. BP: end-to-end backprop, FA: Feedback Alignment, BLL: block local learning, Sim Loss: Local learning with similarity matching loss (Nøkland and Eidnes, 2019). Top-1,3 and 5 accuracies are reported in the respective columns.

four blocks for BLL and Pred-Sim. The splits were introduced after residual layers by grouping subsequent layers into blocks. We also included the predictive loss as suggested in (Nøkland and Eidnes, 2019) in our BLL method (see ablation studies in Supplement to see the role of individual losses in training performance).

Group sizes in the blocks were (4,5,4,5) for ResNet-18 and (12,13,12,13) for ResNet-50. Backward networks for BLL were constructed as linear layers with label size as input and the output size equal to the number of channels in the corresponding ResNet block output. The kernels of ResNet-18/ResNet-50 used by FA architectures during backpropagation were fixed and uniformly initialised following the Kaiming He et al. (2015) initialisation method.

We train ResNet-50 on ImageNet-1K using the standard ImageNet training pipeline from Pytorch (Paszke et al., 2019) without any additional augmentation. We use FFCV (Leclerc et al., 2022) data-loading and training scripts to speed up training. Additional training details and hyperparameters are documented in the Supplement.

The results are summarized in Table 2, top-k test accuracies are shown. Top-3 accuracies count the number of test samples for which the correct class was among the network's three highest output activations (see Supplement for results over multiple runs). BLL performs slightly better than Pred-Sim overall for all tasks and architectures. It also performs close to end-to-end backpropagation performance except for CIFAR-10 using ResNet18 and ImageNet task, hinting at insufficient information being sent to the blocks through the linear feedback network. Unsurprisingly, FA is outperformed by BLL, the gap becoming wider as the task and model complexity increases (Bartunov et al., 2018).

## 4.2 BLOCK-LOCAL TRANSFORMER ARCHITECTURE FOR SEQUENCE-TO-SEQUENCE LEARNING

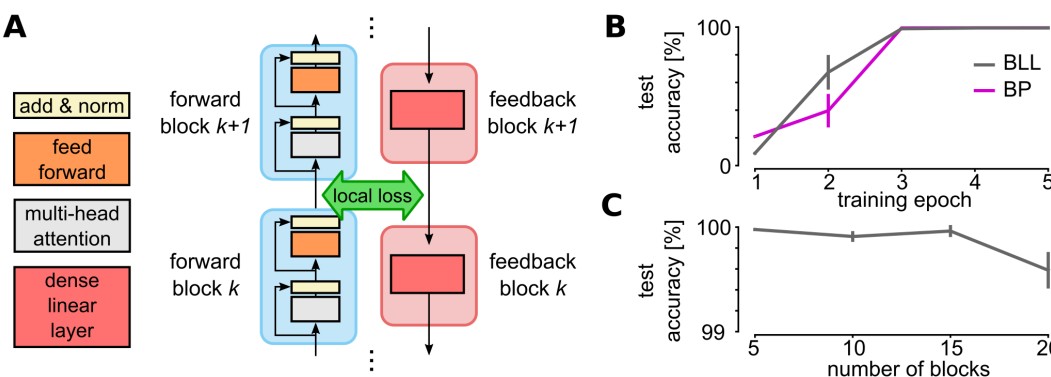

Figure 4: Block local learning of transformer architecture. **A:** Illustration of the transformer forward and feedback network. **B:** Learning curves of block local (BLL) and end-to-end backpropagation (BP) training. **C:** Test accuracy vs. number of blocks in the transformer model. Error bars show standard deviations over 5 runs.

Transformer architectures are well suited for distributed computing due to their repetitive network structure. We demonstrate a proof-of-concept result on training a transformer with BLL. We used a transformer model with 20 self-attention blocks with a single attention head each. Block local losses were added after each block and trained locally. The feedback network was constructed here as a multi-layer network by projecting targets through dense layers and used the local loss for training. See Fig. 4 A for an illustration. The transformer was trained for 5 epochs on a sequence-to-sequence task, where a random permutation of numbers 0..9 was presented and had to be re-generated in reverse order.

BLL achieves a convergence speed that is comparable to that of end-to-end BP on this task. Fig. 4 B shows learning curves of BLL and BP. Both algorithms converge after around 3 epochs to nearly perfect performance. BLL also achieved good performance for a wide range of network depths. Fig. 4 C shows the performance after 5 epochs for different transformer architectures. Using only 5 transformer blocks yields performance of around 99.9% (average over five independent runs). The

test accuracy on this task for the 20 block transformer was 99.6%. These results suggest that BLLod is equally applicable to transformer architectures.

# 5 DISCUSSION

We have demonstrated a probabilistic framework for rigorously defining block-local losses for deep architectures. Our method represents the parameters of probability distributions using the network activations and introduces a feedback network that propagates information backwards from the targets to the input to provide targets for intermediate layers. These targets can be interpreted as prototypical representations that each block must achieve in order to solve the overall classification task. The forward network and the backward feedback can work in parallel and with different sets of weights, solving the locking problem and the weight transport problem. We have shown that our block-local training approach outperforms existing local training approaches and approaches the task performance of backprop in some cases.

While we used linear layers for the feedback network in most of this work, which scales to mid-sized learning problems, in Section 4.2 we demonstrated a proof of concept of using more complex feedback structures. It will be interesting to explore potentially biologically realistic feedback structures for future work as well.

We also showed that our method can scale up to ImageNet and work on different architectures, including transformers. Both of these results on complex tasks and network structures suggest that BLL can scale up to very large models. Our method not only provides a novel way of performing distributed training of large models but also hints at new paradigms of self-supervised training that are biologically plausible.

The proposed method may also help further blur the boundary between deep learning and probabilistic models. Several previous models have shown that DNNs can represent probability distribution (Abdar et al., 2021; Pawlowski et al., 2017; Tran et al., 2019; Malinin and Gales, 2019). Unlike these previous methods, our method does not require Monte Carlo sampling or contrastive training. Instead, it exploits the log-linear structure of exponential family distributions to propagate probabilistic messages efficiently. In fact, we found that combining our local loss with the information-theoretic predictive loss proposed in (Nøkland and Eidnes, 2019) gave the best results. Although BLL was derived using a probabilistic approach, it also shares interesting similarities with earlier non-probabilistic method, such as DirectFeedback Alignment.

Overall, this work addresses an important open problem of modern ML: How can ML models be efficiently distributed and horizontally scaled over many compute nodes for training models too large to fit on one node. Doing so may also allow us to train large models more efficiently, since it would allow us to distribute computation over many smaller, energy-efficient devices rather than a large power-hungry device. This would also make our method especially well suited for new energy efficient hardware for ML, such as neuromorphic devices. The energy consumption and resulting carbon footprint of ML is becoming a major concern and the proposed training method may provide a new direction to reduce the impact of ML.

# REPRODUCIBILITY

We ensure that the results presented in this paper are easily reproducible using just the information provided in the main text as well as the supplement. Details of the models used in our simulations are presented in the main paper and further elaborated in the supplement. We provide additional details and statistics over multiple runs in the supplement section S2. We use publicly available libraries and datasets in our simulations. We will further provide the source code to the reviewers and ACs in an anonymous repository once the discussion forums are opened. This included code will also contain "readme" texts to facilitate easy reproducibility. The theoretical analysis provided in Section 3 is derived in the supplement.

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

# SUPPLEMENTARY INFORMATION

## S1 A PROBABILISTIC FORMULATION OF DISTRIBUTED LEARNING

### S1.1 MARKOV CHAIN MODEL

Here, we provide additional details to the learning model presented in Section 3 of the main text. To establish these results, we consider the Markov chain model $\mathbf{x} \to \mathbf{z}_1 \to \mathbf{z}_2 \to \cdots \to \mathbf{y}$ of a DNN split into $N+1$ blocks, with inputs $\mathbf{x}$, outputs $\mathbf{y}$ and intermediate representations $\mathbf{z}_k$ at block $k$. To simplify the notation we will define the input $\mathbf{z}_0 := \mathbf{x}$ and output $\mathbf{z}_{N+1} := \mathbf{y}$, and $\mathbf{z} = \{\mathbf{z}_k\}, 1 \le k \le N$, the auxiliary latent variables. The DNN suggests a conditional independence structure given by the first-order Markov chain of random variables $\mathbf{z}_k$

$$p\left(\mathbf{y}, \mathbf{z} \mid \mathbf{x}\right) \;=\; p\left(\mathbf{z}_1 \ldots \mathbf{z}_{N+1} \mid \mathbf{z}_0\right) \;=\; \prod_{k=1}^{N+1} p_k\left(\mathbf{z}_k \mid \mathbf{z}_{k-1}, \boldsymbol{\theta}_k\right) \;, \tag{S1}$$

where $p_k\left(\mathbf{z}_k \mid \mathbf{z}_{k-1}, \boldsymbol{\theta}_k\right)$ is the input-output mapping of the $k$-th block subject to block-local network parameters $\boldsymbol{\theta}_k$. If it is clear from the context, we will omit the explicit mention of the parameter vectors $\boldsymbol{\theta}_k$. The computation of messages $\alpha_k$ comes naturally in a feed-forward neural network as the flow of information follows the canonical form, input $\to$ output. Every block of the network thus translates $\alpha_{k-1} \to \alpha_k$ by outputting the statistical parameters of the conditional distribution $p\left(\mathbf{z}_k \mid \mathbf{x}\right)$ and takes $p\left(\mathbf{z}_{k-1} \mid \mathbf{x}\right)$ as input. This interpretation is valid for a suitable split of any DNN into $N+1$ blocks ($N$ is the number of splits) that fulfills a mild set of conditions (see Section S1.4 for details). It is important to note that the random variables $(\mathbf{z}_1, \mathbf{z}_2, \ldots)$ are only implicit. The network generates the parameters to the probability distribution and at no points needs to sample values for these random variables.

### S1.2 USING LATENT REPRESENTATIONS TO CONSTRUCT PROBABILISTIC BLOCK-LOCAL LOSSES

Many commonly used loss functions in deep learning have a probabilistic interpretation, e.g., the cross entropy loss of a binary classifier is identical to the Bernoulli log-likelihood, and the mean squared error corresponds to the log-likelihood of a Gaussian with constant variance. In this formulation, the outputs of the DNN are interpreted as the statistical parameters to a conditional probability distribution (e.g., the mean of a Gaussian) and the loss function measures the support of observed data samples $\mathbf{x}$ and $\mathbf{y}$.

To introduce intermediate block-local representations $\mathbf{z}_k$ in the network, we consider a variational upper bound $\mathcal{F}$ to the log-likelihood loss $\mathcal{L}$

$$\mathcal{F} \;=\; -\log p\left(\mathbf{y} \mid \mathbf{x}\right) + \frac{1}{N} \sum_{k=1}^{N} \mathcal{D}_{KL}\left(q_k \mid p_k\right) \quad \ge \; \mathcal{L} \;, \tag{S2}$$

where $p_k$ and $q_k$ are true and variational posterior distributions over latent variables $p\left(\mathbf{z}_k \mid \mathbf{x}, \mathbf{y}\right)$ and $q\left(\mathbf{z}_k \mid \mathbf{x}, \mathbf{y}\right)$, respectively. Using the Markov property (S1), assuming a fully factorized distribution, implies the conditional independence

$$p\left(\mathbf{y}, \mathbf{z}_k \mid \mathbf{x}\right) = p\left(\mathbf{y} \mid \mathbf{z}_k\right) p\left(\mathbf{z}_k \mid \mathbf{x}\right) \;, \tag{S3}$$

for any $k$. Using this property, we can rewrite Eq. (S2)

$$\mathcal{F} \;=\; -\log p\left(\mathbf{y} \mid \mathbf{x}\right) + \frac{1}{N} \sum_{k=1}^{N} \mathcal{D}_{KL}\left(q_k \mid p_k\right) \;=\; \frac{1}{N} \sum_{k=1}^{N} \mathbb{E}_{q_k}\left[\log \frac{q\left(\mathbf{z}_k \mid \mathbf{x}, \mathbf{y}\right)}{p\left(\mathbf{y}, \mathbf{z}_k \mid \mathbf{x}\right)}\right]$$

$$= \; \frac{1}{N} \sum_{k=1}^{N} \mathbb{E}_{q_k}\left[\log \frac{q\left(\mathbf{z}_k \mid \mathbf{x}, \mathbf{y}\right)}{p\left(\mathbf{z}_k \mid \mathbf{x}\right)} - \log p\left(\mathbf{y} \mid \mathbf{z}_k\right)\right] \;.$$

Finally, the $\frac{1}{N}$ term can be dropped since it is a constant factor with respect to the network parameters, merely scaling the loss and thus ineffective in learning. To arrive at a block-local formulation

of the loss, we separate the generation of the forward and backward messages, and the computation of the local losses. Using this, we write Eq. (S4) in the form

$$\mathcal{F} = \sum_{k=1}^{N} \ell^{(\alpha)}\left(p_k, \beta_k \mid \alpha_{k-1}\right) - \mathbb{E}_{q_k}\left[\log p\left(\mathbf{y} \mid \mathbf{z}_k\right)\right] , \tag{S4}$$

with local losses given by

$$\ell^{(\alpha)}\left(p_k, \beta_k \mid \alpha_{k-1}\right) = \mathcal{D}_{KL}\left(g_k(\alpha_{k-1}, \beta_k) \mid f_k(\alpha_{k-1})\right) = \mathcal{D}_{KL}\left(q_k \mid \alpha_k\right) ,$$

where we defined the mapping $f_k(\alpha_{k-1}) = \mathbb{E}_{\alpha_{k-1}}\left[p_k\left(\mathbf{z}_k \mid \mathbf{z}_{k-1}\right)\right] = \alpha_k$ and $g_k(\alpha_{k-1}, \beta_k) = \mathbb{E}_{\alpha_{k-1}}\left[\frac{1}{\mathcal{Z}} p_k\left(\mathbf{z}_k \mid \mathbf{z}_{k-1}\right) \beta(\mathbf{z}_k, \mathbf{y})\right] = \mathbb{E}_{\alpha_{k-1}}\left[q\left(\mathbf{z}_k \mid \mathbf{z}_{k-1}, \mathbf{y}\right)\right) = q_k$, with normalization $\mathcal{Z}$. Eq. (S4) is an upper bound on the log-likelihood loss $\mathcal{L} = -\log p\left(\mathbf{y} \mid \mathbf{x}\right) \leq \mathcal{F}$. Since $\mathcal{L}$ is strictly positive, minimizing $\mathcal{F}$ to zeros implies that also $\mathcal{L}$ becomes zero (Jordan et al., 1999). We can also add any positive auxiliary loss $\tilde{\ell}\left(p_k, \beta_k \mid \alpha_{k-1}\right) \geq 0$ to $\ell^{(\alpha)}$, which results in a new upper bound for $\mathcal{L}$. Therefore, in Eq. (S4) we can also use the augmented loss

$$\ell\left(p_k, \beta_k \mid \alpha_{k-1}\right) = \mathcal{D}_{KL}\left(q_k \mid \alpha_k\right) + \tilde{\ell}\left(p_k, \beta_k \mid \alpha_{k-1}\right) , \tag{S5}$$

instead of $\ell^{(\alpha)}\left(p_k, \beta_k \mid \alpha_{k-1}\right)$ directly. Importantly, all terms of the loss $\ell$ can be computed locally through the forward propagation of the $k$-th block to realize $f_k$ and the computation of the posterior $g_k$.

The variational posterior $q$ is given by Eq. (4). Alternatively we can also use a multi-layer feedback network, that propagates messages $\beta_k$ backward

$$\begin{aligned}
\beta_k\left(\mathbf{z}_k\right) = p\left(\mathbf{y} \mid \mathbf{z}_k\right) &= \mathbb{E}_{\mathbf{z}_{k+1}}\left[p_k\left(\mathbf{y} \mid \mathbf{z}_{k+1}\right) p_k\left(\mathbf{z}_{k+1} \mid \mathbf{z}_k\right)\right] \\
&= \mathbb{E}_{\mathbf{z}_{k+1}}\left[\beta_{k+1}\left(\mathbf{z}_{k+1}\right) p_k\left(\mathbf{z}_{k+1} \mid \mathbf{z}_k\right)\right] .
\end{aligned} \tag{S6}$$

This is the method that was using in Section 4.2. This method re-introduces a locking problem but may work better in some scenarios where more complex feedback messages are required. Also here the feedback cannot be computed in closed form so we resort to a gradient-based method.

### S1.2.1 ESTIMATING THE LOG-LIKELIHOOD LOSS THROUGH POSTERIOR BOOTSTRAPPING

Next, we show how the remaining term $\mathbb{E}_{q_k}\left[\log p\left(\mathbf{y} \mid \mathbf{z}_k\right)\right]$ in Eq. (S4) can be estimated locally. The intuition behind this result is that $-\mathbb{E}_{q_k}\left[\log p\left(\mathbf{y} \mid \mathbf{z}_k\right)\right]$ is of a similar form as the log-likelihood loss (Eq. (1) of the main text), i.e., the likelihood of the data labels $\mathbf{y}$ of the residual network $\mathbf{z}_k \rightarrow \mathbf{y}$. Thus, treating $\mathbf{z}_k$ as block-local input data and minimizing the augmented ELBO loss from layer $\mathbf{z}_k \rightarrow \mathbf{y}$ minimizes another upper bound on the global loss $\mathcal{L}$. To formalize this observation we introduce the recursive short-hand notation $q_{k \rightarrow j} = \mathbb{E}_{q_{k \rightarrow (j-1)}}\left[q\left(\mathbf{z}_j \mid \mathbf{z}_{j-1}, \mathbf{y}\right)\right]$, with $q_{k \rightarrow k} = q_k$. Using this, we find the following chain of inequalities

$$-\mathbb{E}_{q_k}\left[\log p\left(\mathbf{y} \mid \mathbf{z}_k\right)\right]$$
$$\leq -\mathbb{E}_{q_k}\left[\log p\left(\mathbf{y} \mid \mathbf{z}_k\right)\right] + \mathcal{D}_{KL}\left(\mathbb{E}_{q_k}\left[q\left(\mathbf{z}_{k+1} \mid \mathbf{z}_k, \mathbf{y}\right)\right] \mid \mathbb{E}_{q_k}\left[p\left(\mathbf{z}_{k+1} \mid \mathbf{z}_k, \mathbf{y}\right)\right]\right) \tag{S7}$$

$$\leq \mathbb{E}_{q_{k \rightarrow (k+1)}}\left[\log \mathbb{E}_{q_k}\left[q\left(\mathbf{z}_{k+1} \mid \mathbf{z}_k, \mathbf{y}\right)\right] - \mathbb{E}_{q_k}\left[\log p\left(\mathbf{z}_{k+1} \mid \mathbf{z}_k, \mathbf{y}\right) + \log p\left(\mathbf{y} \mid \mathbf{z}_k\right)\right]\right] \tag{S8}$$

$$= \mathbb{E}_{q_{k \rightarrow (k+1)}}\left[\log \mathbb{E}_{q_k}\left[q\left(\mathbf{z}_{k+1} \mid \mathbf{z}_k, \mathbf{y}\right)\right] - \mathbb{E}_{q_k}\left[\log p_{k+1}\left(\mathbf{z}_{k+1} \mid \mathbf{z}_k\right)\right]\right] - \mathbb{E}_{q_{k \rightarrow (k+1)}}\left[\log p\left(\mathbf{y} \mid \mathbf{z}_{k+1}\right)\right]$$

$$\leq \mathcal{D}_{KL}\left(\mathbb{E}_{q_k}\left[q\left(\mathbf{z}_{k+1} \mid \mathbf{z}_k, \mathbf{y}\right)\right] \mid \mathbb{E}_{q_k}\left[p_{k+1}\left(\mathbf{z}_{k+1} \mid \mathbf{z}_k\right)\right]\right)$$
$$\quad - \mathbb{E}_{q_k, q_{k \rightarrow (k+1)}}\left[\log p_{k+1}\left(\mathbf{z}_{k+1} \mid \mathbf{z}_k\right)\right] - \mathbb{E}_{q_{k \rightarrow (k+1)}}\left[\log p\left(\mathbf{y} \mid \mathbf{z}_{k+1}\right)\right] \tag{S9}$$

$$= \ell_{k,k+1}^{(\rho)} - \mathbb{E}_{q_{k \rightarrow (k+1)}}\left[\log p\left(\mathbf{y} \mid \mathbf{z}_{k+1}\right)\right] , \tag{S10}$$

with local loss

$$\ell_{k,l}^{(\rho)} = \mathcal{D}_{KL}\left(\mathbb{E}_{q_{k \rightarrow l}}\left[q\left(\mathbf{z}_{l+1} \mid \mathbf{z}_l, \mathbf{y}\right)\right] \mid \mathbb{E}_{q_{k \rightarrow l}}\left[p_{l+1}\left(\mathbf{z}_{l+1} \mid \mathbf{z}_l\right)\right]\right) - \mathcal{H}\left(p_{l+1} \mid q_{k \rightarrow l}\right)$$
$$\text{with} \quad \mathcal{H}\left(p_{l+1} \mid q_{k \rightarrow l}\right) = \mathbb{E}_{q_{k \rightarrow l}, q_{k \rightarrow (l+1)}}\left[\log p_{l+1}\left(\mathbf{z}_{l+1} \mid \mathbf{z}_l\right)\right] \tag{S11}$$

and where $\mathbb{E}_{q_{k \rightarrow l}}\left[\right]$ denotes expectation with respect to $q_{k \rightarrow l}$. In (S8) we used Jensen's inequality, $-\mathbb{E}\left[\log p\left(X\right)\right] \geq -\log \mathbb{E}\left[p\left(X\right)\right]$, and in (S9) we used $-\log \mathbb{E}_{q_k}\left[\alpha_{k+1}(\mathbf{z}_k)\right] \geq 0$, i.e., the negative log expectation of a probability distribution is always positive or zero.

Next we generalize the inequality (S9) in the following theorem:

**Theorem S1.1.** *Let $p\left(\mathbf{y}, \mathbf{z} \mid \mathbf{x}\right)$ be a probabilistic model subject to the conditional independence properties over $N + 1$ blocks, given by Eq. (S1). Let $\ell_k^{(\rho)}$ be the local loss function Eq.(S11). Furthermore, let $\ell_k^{(\omega)} = -\mathbb{E}_{q_{k \to N}}\left[\log p\left(\mathbf{y} \mid \mathbf{z}_N\right)\right]$. Then the log-likelihood loss Eq. (1) is bounded from above by the sum of losses $\mathcal{F}_N \geq \mathcal{L}$*

$$\mathcal{F}_N = \sum_{k=1}^{N} \ell_k^{(\alpha)} + \ell_k^{(\omega)} + \sum_{l=k+1}^{N} \ell_{k,l}^{(\rho)} \tag{S12}$$

*Proof.* We prove Theorem 1 by induction over block $i$. Let $\mathcal{L} = -\log p\left(\mathbf{y} \mid \mathbf{x}\right)$, and $\mathcal{F}_1 = \mathcal{F}$ as in Eq. (S2). We show a transition $\mathcal{F}_{i-1} \to \mathcal{F}_i$, with $\mathcal{F}_{i-1} \leq \mathcal{F}_i$, and where $\mathcal{F}_N$ recovers Eq. (S12). This result implies a hierarchy of loss functions $0 \leq \mathcal{L} \leq \mathcal{F}_1 \leq \mathcal{F}_2 \leq ... \leq \mathcal{F}_N$, and $\mathcal{L} \leq \mathcal{F}_N$ follows, which completes the proof.

We define:

$$\mathcal{F}_i = \sum_{k=1}^{N} \ell_k^{(\alpha)} + \sum_{l=k+1}^{i} \ell_{k,l}^{(\rho)} - \mathbb{E}_{q_{k \to j}}\left[\log p\left(\mathbf{y} \mid \mathbf{z}_j\right)\right] \Bigg|_{j=max(i,k)} \tag{S13}$$

For $i = 1$ we recover Eq. (S4) and thus $\mathcal{L} \leq \mathcal{F}_1$ holds as established before. Using the result (S9) we can take the inductive step $\mathcal{L}_{i-1} \to \mathcal{L}_i$

$$\begin{aligned}
\mathcal{F}_{i-1} &= \sum_{k=1}^{N} \ell_k^{(\alpha)} + \sum_{l=k+1}^{i-1} \ell_{k,l}^{(\rho)} - \mathbb{E}_{q_{k \to j}}\left[\log p\left(\mathbf{y} \mid \mathbf{z}_j\right)\right] \Bigg|_{j=max(i-1,k)} \\
&= \sum_{k=1}^{i-1} \ell_k^{(\alpha)} + \sum_{l=k+1}^{i-1} \ell_{k,l}^{(\rho)} - \mathbb{E}_{q_{k \to (i-1)}}\left[\log p\left(\mathbf{y} \mid \mathbf{z}_{i-1}\right)\right] + \sum_{k'=i}^{N} \ell_{k'}^{(\alpha)} - \mathbb{E}_{q_{k'}}\left[\log p\left(\mathbf{y} \mid \mathbf{z}_{k'}\right)\right] \\
&\leq \sum_{k=1}^{i-1} \ell_k^{(\alpha)} + \sum_{l=k+1}^{i-1} \ell_{k,l}^{(\rho)} + \ell_{k,i}^{(\rho)} - \mathbb{E}_{q_{k \to i}}\left[\log p\left(\mathbf{y} \mid \mathbf{z}_i\right)\right] \\
&\quad + \ell_i^{(\alpha)} + \ell_{i,i+1}^{(\rho)} - \mathbb{E}_{q_{i \to (i+1)}}\left[\log p\left(\mathbf{y} \mid \mathbf{z}_{i+1}\right)\right] + \sum_{k'=i+1}^{N} \ell_{k'}^{(\alpha)} - \mathbb{E}_{q_{k'}}\left[\log p\left(\mathbf{y} \mid \mathbf{z}_{k'}\right)\right] \\
&= \sum_{k=1}^{N} \ell_k^{(\alpha)} + \sum_{l=k+1}^{i} \ell_{k,l}^{(\rho)} - \mathbb{E}_{q_{k \to j}}\left[\log p\left(\mathbf{y} \mid \mathbf{z}_j\right)\right] \Bigg|_{j=max(i,k)} = \mathcal{F}_i . \tag{S14}
\end{aligned}$$

This shows that the global loss can be decomposed into a sum of local losses. Setting $i = N$ in Eq. (S13), the proof of Theorem S1.1 follows. $\qquad\square$

What remains to be shown is that posterior bootstrapping allows us to compute the required terms. To arrive at this result we further study the local loss (S11). Note, that this expression can be written in the form (S5) as a function of the forward network transfer $p\left(\mathbf{z}_{l+1} \mid \mathbf{z}_l\right)$ given the distribution $q_{k \to l}(\mathbf{z}_l)$ for $1 \leq k \leq l \leq N$, i.e.

$$\begin{aligned}
\ell_{k,l}^{(\rho)} &= \ell\left(p_{l+1}, \beta_{l+1} \mid q_{k \to l}\right) = \\
&\quad \mathcal{D}_{KL}\left(g_k(q_{k \to l}, \beta_{l+1}) \mid f_k(q_{k \to l})\right) - \mathcal{H}\left(p_{l+1} \mid q_{k \to l}\right) , \tag{S15}
\end{aligned}$$

where we used the mappings $f_k$ and $g_k$ as in Eq. (S5). Thus by choosing $\hat{\ell}\left(p_{l+1}, \beta_{l+1} \mid q_{k \to l}\right) = \mathcal{H}\left(p_{l+1} \mid q_{k \to l}\right)$, we can spell out the local loss in the exact same way as Eq. (S5), but passing the posterior messages $q_{k \to l}$ instead of the forward pass $\alpha_k$. The last block with index $N + 1$ directly optimizes $\ell_k^{(\omega)} = \mathbb{E}_{q_{k \to N}}\left[\log p\left(\mathbf{y} \mid \mathbf{z}_N\right)\right]$, which is also local to that block.

We thus propose to realize the sum over $k, l$ in (S14) using a posterior bootstrapping schedule. Instead of passing only the forward messages, blocks may be selected to compute the variational posterior distribution $q$ locally and pass that to the next block instead. The optimal posterior bootstrapping schedule, according to (S14), is the one that computes all $N^2$ combinations of passing $\alpha$ and $q$ messages giving rise to the structure in Fig. 2. We are now ready to prove Theorem 1, which we reverberate here for completeness

**Theorem S1.2.** *Let $p(\mathbf{y}, \mathbf{z} \,|\, \mathbf{x})$ be a probabilistic model subject to the conditional independence properties over $N+1$ blocks, given by Eq. (S1). Let $\ell$ be the local loss function Eq.(S5). Furthermore, let $\alpha_k^{(m)}$, $\beta_k^{(m)}$ be messages created using the optimal posterior bootstrapping schedule outlined above. Then, the simultaneous minimization of $\ell\left(p_l^{(m)}, \beta_l^{(m)} \,\middle|\, \alpha_k^{(m)}\right)$ in all blocks $k$, minimizes an upper bound on the log-likelihood loss $\mathcal{L}$.*

*Proof.* The proof of Theorem S1.2 follows directly from Theorem S1.1 by substituting the loss terms in the sums with (S15). Importantly, all messages generated by the bootstrapping schedule are treated the same, so there is no contrastive step or need for global information to signal a network-wide learning phase. In simulations we also experimented with different bootstrapping schedules, other than the optimal one. $\qquad\square$

### S1.3 RELATIONSHIP TO EM AND CONVERGENCE PROPERTIES

As outlined above the model can be closely linked to the EM algorithm. The split of gradient estimators using the Markov assumption is a key property of algorithms derived from EM, and also the key property exploited in BLL. EM makes use of the identity (Dempster et al., 1977)

$$
\begin{aligned}
-\nabla\mathcal{L} \;=\; \nabla \log p(\mathbf{y} \,|\, \mathbf{x}) &\;=\; \frac{1}{p(\mathbf{y} \,|\, \mathbf{x})} \nabla p(\mathbf{y} \,|\, \mathbf{x}) \\
&\;=\; \frac{1}{p(\mathbf{y} \,|\, \mathbf{x})} \nabla \mathbb{E}_{\mathbf{z}_k}\left[p(\mathbf{y} \,|\, \mathbf{z}_k)\, p(\mathbf{z}_k \,|\, \mathbf{x})\right] \;=\; \mathbb{E}_{p(\mathbf{z}_k \,|\, \mathbf{x},\mathbf{y})}\left[\nabla \log p(\mathbf{y} \,|\, \mathbf{z}_k) + \nabla \log p(\mathbf{z}_k \,|\, \mathbf{x})\right],
\end{aligned}
$$

where in the last step we used that $p(\mathbf{y} \,|\, \mathbf{x})$ is constant under the expectation and $\frac{p(\mathbf{y} \,|\, \mathbf{z}_k)\, p(\mathbf{z}_k \,|\, \mathbf{x})}{p(\mathbf{y} \,|\, \mathbf{x})} = p(\mathbf{z}_k \,|\, \mathbf{x}, \mathbf{y})$ (Bayes' rule).

We use a variational approach where the posterior is replaced by $q$. It has been established in prior work that, similar to the EM algorithm, the variational loss $\mathcal{L}$ can be minimized by alternating two optimization steps (Jordan et al., 1999; Neal and Hinton, 1998)

$$
\text{E-step:}\quad q^{(t)} \;=\; \arg\min_q \mathcal{F}\left(q, \boldsymbol{\theta}^{(t-1)}\right) \tag{S16}
$$

$$
\text{M-step:}\quad \boldsymbol{\theta}^{(t)} \;=\; \arg\min_{\boldsymbol{\theta}} \mathcal{F}\left(q^{(t)}, \boldsymbol{\theta}\right). \tag{S17}
$$

In Neal and Hinton (1998) it was shown that this approach also works if gradient descent is used for the optimization of (some of) the parameters. We use here the variant where parameters of the forward network are optimized via gradient descent whereas the loss with respect to the feedback network parameters is directly optimized.

### S1.4 GENERAL EXPONENTIAL FAMILY DISTRIBUTION

To arrive at a result for the gradient of the first (KL-divergence) term $\ell_k$ in Eq. (S4) we seek distributions for which the marginals can be computed in closed form. We assume forward messages $\alpha$ and posterior $\rho$ be given by general exponential family distributions

$$
\alpha_k(\mathbf{z}_k) \;=\; \prod_j \alpha_{kj}(z_{kj}) \;=\; \prod_j h(z_{kj})\exp\left(T(z_{kj})\,\phi_{kj} - A(\phi_{kj})\right) \tag{S18}
$$

$$
\rho_k(\mathbf{z}_k) \;=\; \prod_j \rho_{kj}(z_{kj}) \;=\; \prod_j h(z_{kj})\exp\left(T(z_{kj})\,\gamma_{kj} - A(\gamma_{kj})\right) \tag{S19}
$$

with base measure $h$, sufficient statistics $T$, log-partition function $A$, and natural parameters $\phi_{kj}$ and $\gamma_{kj}$. Using this the KL loss becomes

$$
\mathcal{D}_{KL}(\rho_k \,|\, \alpha_k) \;=\; \sum_j \mathbb{E}_{\rho_{kj}}\left[T(z_{kj})(\phi_{kj} - \gamma_{kj}) - A(\phi_{kj}) + A(\gamma_{kj})\right], \tag{S20}
$$

and thus

$$
-\nabla\mathcal{D}_{KL}\left(\rho_k\,|\,\alpha_k\right) \;=\; \sum_j \left(\mathbb{E}_{\rho_{kj}}\left[T\left(z_{kj}\right)\right] - \mathbb{E}_{\alpha_{kj}}\left[T\left(z_{kj}\right)\right]\right)\nabla\phi_{kj} \;+
$$

$$
\underbrace{\left(\mathbb{E}_{\rho_{kj}}\left[T\left(z_{kj}\right)^2\right] - \mathbb{E}_{\rho_{kj}}\left[T\left(z_{kj}\right)\right]^2\right)}_{\sigma^2(\rho_{kj})}\left(\phi_{kj} - \gamma_{kj}\right)\nabla\gamma_{kj}\;, \qquad \text{(S21)}
$$

which by defining $\mu\left(p\right) = \mathbb{E}_p\left[T\left(z_{kj}\right)\right]$ can be written in the compact form

$$
-\nabla\mathcal{D}_{KL}\left(\rho_k\,|\,\alpha_k\right) \;=\; \sum_j \left(\mu\left(\rho_{kj}\right) - \mu\left(\alpha_{kj}\right)\right)\nabla\phi_{kj} \;+\; \sigma^2\left(\rho_{kj}\right)\left(\phi_{kj} - \gamma_{kj}\right)\nabla\gamma_{kj}\;.
$$

This is the result Eq. (6) of the main text.

### S1.4.1  GAUSSIAN RANDOM VARIABLES WITH KNOWN VARIANCE

Throughout the numerical simulations we use the network to represent Gaussian distributions with known variance. For this distribution we have $T\left(z_{kj}\right) = z_{kj}$, $\mathbb{E}_{\rho_{kj}}\left[T\left(z_{kj}\right)\right] = \phi_{kj}$, and furthermore $\sigma^2\left(\rho_{kj}\right) = \sigma^2\left(= const\right)$. We get

$$
-\nabla\ell_k \;=\; \sum_j \left(\gamma_{kj} - \phi_{kj}\right)\nabla\phi_{kj} \;+\; \sigma\left(\phi_{kj} - \gamma_{kj}\right)\nabla\gamma_{kj} \qquad \text{(S22)}
$$

Using the parameterization $\phi_{kj} = a_{kj}$ and $\gamma_{kj} = \frac{1}{2}\left(a_{kj} + b_{kj}\right)$, we further get

$$
-\nabla_a\,\ell_k \;=\; \left(\frac{\sigma}{2} - 1\right)\sum_j \left(a_{kj} - b_{kj}\right)\nabla a_{kj}\;. \qquad \text{(S23)}
$$

This is the KL loss that was used to minimize the distance between forward and feedback features.

### S1.5  CLOSED FORM SOLUTION OF BACKWARD NETWORK

Here we show the closed form solution for optimizing the backward network. Over a set of $M$ training samples we seek to solve

$$
\boldsymbol{\theta}_k^{(b)} \;\overset{!}{=}\; \underset{\boldsymbol{\theta}_k^{(b)}}{\arg\min}\; \sum_{m=1}^{M} \ell_k(\rho_k^{(m)}, \alpha_k^{(m)}) \;=\; \underset{\theta_k^{(b)}}{\arg\min}\; \sum_{m=1}^{M} \mathcal{D}_{KL}\left(\rho_k^{(m)}\,\Big|\,\alpha_k^{(m)}\right) \;+\; \mathcal{H}\left(\rho_k^{(m)}, \alpha_k^{(m)}\right)\;.
$$

As in the remainder of this paper, we treat the inputs to the block $k$ as constants. The second cross-entropy term only depends on the parameters of the forward network. Taking the gradient with respect to backward network parameters $\gamma_{kj}$ thus yields

$$
\nabla_{\gamma_{kj}}\,\ell_k = \nabla_{\gamma_{kj}}\mathcal{D}_{KL}\left(\rho_k\,|\,\alpha_k\right)
$$

$$
= \sum_{m=1}^{M} \nabla\mu\left(\gamma_{kj}^{(m)}\right)\left(\gamma_{kj}^{(m)} - \phi_{kj}^{(m)}\right)\nabla\gamma_{kj}^{(m)} + \mu\left(\gamma_{kj}^{(m)}\right)\nabla\gamma_{kj}^{(m)} - \nabla A\left(\gamma_{kj}^{(m)}\right)\nabla\gamma_{kj}^{(m)} \overset{!}{=} 0
$$

$$
\leftrightarrow \quad \sum_{m=1}^{M} \nabla\mu\left(\gamma_{kj}^{(m)}\right)\left(\gamma_{kj}^{(m)} - \phi_{kj}^{(m)}\right) + \mu\left(\gamma_{kj}^{(m)}\right) - \nabla A\left(\gamma_{kj}^{(m)}\right) \overset{!}{=} 0
$$

Assuming Gaussian with known variance $\mu\left(\gamma_{kj}^{(m)}\right) = \sigma\,\gamma_{kj}^{(m)}$, $\nabla\mu\left(\gamma_{kj}^{(m)}\right) = \sigma$, $A\left(\gamma_{kj}^{(m)}\right) = \frac{\left(\gamma_{kj}^{(m)}\right)^2}{2}$ and $\nabla A\left(\gamma_{kj}^{(m)}\right) = \gamma_{kj}^{(m)}$ gradient with respect to $\gamma_{kj}$

$$
\leftrightarrow \quad \sum_{m=1}^{M} \nabla\mu\left(\gamma_{kj}^{(m)}\right)\left(\gamma_{kj}^{(m)} - \phi_{kj}^{(m)}\right) + \mu\left(\gamma_{kj}^{(m)}\right) - \nabla A\left(\gamma_{kj}^{(m)}\right) \overset{!}{=} 0
$$

$$
\leftrightarrow \quad \sum_{m=1}^{M} \sigma\left(\gamma_{kj}^{(m)} - \phi_{kj}^{(m)}\right) + \sigma\,\gamma_{kj}^{(m)} - \gamma_{kj}^{(m)} \overset{!}{=} 0
$$

$$
\leftrightarrow \quad \sum_{m=1}^{M} \gamma_{kj}^{(m)}\left(2\,\sigma - 1\right) - \sigma\phi_{kj}^{(m)} \overset{!}{=} 0
$$

| Hyperparameter | Value |
|---|---|
| weight of KL loss | 0.70 |
| weight of entropy loss $\mathcal{H}$ | 0.014 |
| magnitude of added noise to estimate $\mathcal{H}$ | 0.013 |
| weight of correlation loss | 0.70 |
| predictive loss scaling | 0.1 |
| weight of output CE loss | 0.49 |
| posterior bootstrapping | optimal |
| batch size | 256 |

Table S1: Hyperparameters used for training ResNet-50 on FashionMNIST task.

$$\gamma_{kj}^{(m)} = \tfrac{1}{2}\left(a_{kj}^{(m)} + b_{kj}\right)$$

$$\leftrightarrow \quad \sum_{m=1}^{M} \frac{1}{2}\left(a_{kj}^{(m)} + b_{kj}\right)(2\,\sigma - 1) - \sigma a_{kj}^{(m)} \stackrel{!}{=} 0$$

$$\leftrightarrow \quad \frac{M}{2}\,(2\,\sigma - 1)\,b_{kj} - \frac{1}{2}\sum_{m=1}^{M} a_{kj}^{(m)} \stackrel{!}{=} 0$$

$$\leftrightarrow \quad b_{kj} \stackrel{!}{=} \frac{1}{M\,(2\,\sigma - 1)}\sum_{m=1}^{M} a_{kj}^{(m)} = \frac{c_1}{M}\sum_{m=1}^{M} a_{kj}^{(m)},$$

with constant $c_1$. The optimal parameters for the backward network is thus given by the class-specific mean over the forward messages.

## S2    NUMERICAL SIMULATIONS

We assessed the models results variability over five runs for each model and each task, using different random seeds. We used 5 different losses derived from our theoretical framework or previously established: the BLL *KL loss*, the *entropy loss $\mathcal{H}$*, the *prediction loss* as in (Nøkland and Eidnes, 2019), a *correlation loss* that punishes high auto-correlation of features within a batch and the output cross-entropy (CE) loss, that is only effective at the last block. Each loss was assigned a weight to scale it relative to the other losses and then combined to block-local losses that were optimized individually.

### S2.0.1    FASHIONMNIST CLASSIFICATION TASK

FashionMNIST is a freely available dataset consisting of 60k training grayscale images and 10k grayscale test images of fashion items published under the MIT License (MIT) (Xiao et al., 2017). The images were normalized to have mean 0 and stds 1 and augmented with random horizontal flips during training. The BLL networks for FashionMNIST experiments used the same ResNet architectures but augmented with the feedback blocks. For the forward network we used the Adam optimizer with a learning rate of 0.03 without weight decay, a Cosine annealing learning rate (LR) scheduler (Loshchilov and Hutter, 2017) with max iterations set to 140. We used the direct closed form optimization for the feedback network on every batch but applied it with a rate of only 0.9 to account for missing classes. The remaining hyperparameters used are given in Table S1.

### S2.0.2    CIFAR10 CLASSIFICATION TASK

CIFAR10 is a freely available dataset consisting of 50k training images and 10k test images from (Krizhevsky, 2009). We used the same data augmentation, optimizers and hyperparameters used for FashionMNIST to train CIFAR10 (see Table S1).

|  |  | test-1 (mean±std) | test-3 (mean±std) |
|---|---|---|---|
| ResNet-18 | BLL | 94±0.2 | 99.3±0.04 |
|  | BP | 92.7±0.1 | 99.2±0.7 |
|  | FA | 88.2±0.3 | 98.7±0.2 |
|  | Pred-Sim | 93.7 ±0.2 | 99.4±0.1 |
| ResNet-50 | BLL | 94.1±0.24 | 99.1±0.1 |
|  | BP | 93.4±0.6 | 99.4±0.05 |
|  | FA | 86.6±0.7 | 98.6±0.1 |
|  | Pred-Sim | 94.2 ±0.2 | 99.4±0.08 |

Table S2: As in Table 2. Classification accuracy (% correct) for 5 runs on FashionMNIST vision tasks.

|  |  | test-1 (mean±std) | test-3 (mean±std) |
|---|---|---|---|
| ResNet-18 | BLL | 88.1±0.2 | 97.9±0.1 |
|  | BP | 92.5±1.5 | 98.3±0.3 |
|  | FA | 72.0±0.6 | 92.8±0.1 |
|  | Pred-Sim | 87.8 ±0.2 | 97.9±0.1 |
| ResNet-50 | BLL | 92.3±0.2 | 98.9±0.1 |
|  | BP | 91.1±1.1 | 98.7±0.2 |
|  | FA | 62.5±0.4 | 88.2±0.2 |
|  | Pred-Sim | 92.1 ±0.2 | 98.8±0.1 |

Table S3: As in Table 2. Classification accuracy (% correct) for 5 runs on CIFAR10 task.

## S2.1 IMAGENET CLASSIFICATION TASK

We train ResNet-50 on Imagenet-1K using FFCV (Leclerc et al., 2022) library. Standard hyperparameters of ImageNet were not changed from the FFCV baseline results. We reused most of the hyperparameters specific to our method from the CIFAR-10 task. See Table S4 for full details on the hyperparameters used for Imagenet-1K.

## S2.2 ABLATION STUDY

We performed ablation studies to assess the importance of the different losses in BLL: Correlation loss, predictive loss and KL loss. To this end, we disabled one loss at a time and trained on CIFAR-10. In addition we also disabled all local losses, effectively only training the last block directly at the

| Hyperparameter | Value |
|---|---|
| weight of KL loss | 0.25 |
| weight of entropy loss $\mathcal{H}$ | 1.0 |
| magnitude of added noise to estimate $\mathcal{H}$ | 0.01 |
| weight of correlation loss | 0.7 |
| feedback network LR | 0.9 |
| weight of output CE loss | 0.43 |
| predictive loss scaling | 0.5 |
| posterior bootstrapping | disabled |
| batch size | 512 |
| feedback optimizer momentum | 0.01 |

Table S4: Hyperparameters used for training ResNet-50 on ImageNet task. All other hyperparameters relating to theforward network training are not modified from the baseline FFCV training script

| modification | performance |
|---|---|
| w/o correlation loss | 91.4±0.2 |
| w/o predictive loss | 91.5±0.2 |
| w/o KL loss | 91.8±0.3 |
| w/o all local losses | 52.1±13.3 |
| with simplified bootstrapping | 92.4±0.1 |
| benchmark BLL | 92.3±0.2 |

Table S5: BLL CIFAR-10 test accuracy with modified algorithm

targets. Furthermore, we assess the importance of the optimal posterior bootstrapping by using only the simplified bootstrapping schedule where only forward messages are propagated while keeping all losses enabled. The results are presented in Table S5.

A decrease in performance is observed whenever one of the local losses is removed, but removing them all drastically reduces the performance as expected. No local losses means training only the block layer while freezing the remaining blocks, thus the decrease in performance. Using a simplified boostrapping scheme gives comparable performance as augmenting forward messages with backward messages. In this case the message augmentation doesn't provide sensible advantage. Exploring different augmentation methods on more tasks and architectures might give better insight.

We study the effect of splitting the network into blocks in Figure S1. The performance decreases as the number introduced in the network increases. This effect is more pronounced as the difficulty of the task increases. We compare this to the effect of adding additional losses to the training method without splitting the network. This network is trained end-to-end with backpropagation, these additional losses introduced slight performance degradation.

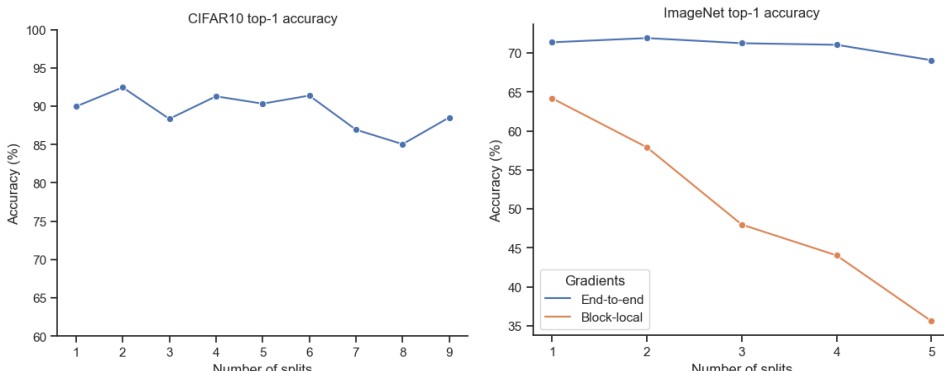

Figure S1: Top-1 classification accuracy on CIFAR-10 (Left) and ImageNet (Right) across number of splits in the ResNet-50. ImageNet performance at 30 epochs of training is compared to backpropagation training while keeping additional losses introduced in our method.

## S2.3 HARDWARE AND SOFTWARE DETAILS

ResNet18 and ResNet50 models and experiments were implemented in PyTorch (Paszke et al., 2019). Most of our experiments were run on NVIDIA A100 GPUs and some initial evaluations and the MINST experiments were conducted on NVIDIA V100 and Quadro RTX 5000 GPUs. In total we used about 190,000 core hours for training and hyper-parameter searches.

