# OpenReview forum: "Block-local learning with probabilistic latent representations"
_ICLR.cc/2024/Conference — Submitted to ICLR 2024_

### Official Review · Reviewer_ACie · 2023-10-29

**Soundness:** 3 good
**Presentation:** 3 good
**Contribution:** 4 excellent
**Rating:** 5
**Confidence:** 2

**Summary:**

In this paper, a new training method is proposed for neural network architectures. Also, the authors provide a novel theoretical framework to analyze deep neural networks as parameters of probability distributions. Based on it, a new training method is proposed, with theory and numerical experiments on classification tasks.

**Strengths:**

The theoretical analysis to interpret DNN for probability distribution viewpoint is interesting. Also, the paper is well organized and well written to prove the new concept and method.

**Weaknesses:**

Please see the questions in the following part. There are some details not clear enough. Also, the numerical experiment results are not satisfying.

**Questions:**

Here are some questions for this paper:
Q: How to the principle or theory to select split blocks for each architecture?
Q: How to deal with models with multiple branches? Is it possible to split the network for each branch?
Q: What is the reason for the large performance gap on Cifar10 and ImageNet tasks, compare with BP method? Are there using same data preprocessing and augmentations?
Q: What is the training time and convergence speed compared with BP method?

---

> ### Author Response · Authors · 2023-11-21
>
> *Q: How to the principle or theory to select split blocks for each architecture?*
>
> In general, the number of blocks will be determined based on the requirements of distributing the model (model size, no. of devices, memory per device etc.). There is a trade-off in number of splits and sacrifice in performance which depends on learning task and network architecture (See Figure S1).
>
> *Q: How to deal with models with multiple branches? Is it possible to split the network for each branch?*
>
> We did not test models with multiple branches yet. However, the theory would support such models. Will be interesting to test that in future work.
>
> *Q: What is the reason for the large performance gap on Cifar10 and ImageNet tasks, compare with BP method? Are there using same data preprocessing and augmentations?*
>
> Yes, they use the same preprocessing and augmentation. We think there are two main reasons for the gap: 1) due to time and resource limitations we were not able to do any hyper-parameter search for ImageNet. 2) For tasks with number of classes in the range of the batch size, the greedy update of the backward network may not be suitable (backward messages are estimated by 1-2 samples on expectation which doesn't provide rich enough statistics). Non-greedy updates of backward nets would likely fix that, which we are currently developing. We still thought it is beneficial for the paper to include the ImageNet experiments in the current, since the performance is at least not trivial. We will update the discussion and clarify these shortcomings of the current form of the algorithm.
>
> *Q: What is the training time and convergence speed compared with BP method?*
>
> We implemented 2 versions of our algorithm, one a naive implementation which is as expensive as backprop (with additional linear layers for the backward network) and second , as the reviewer rightly suggested to put different blocks on multiple devices. However the updates are still locked due to pytorch’s limitations. We consider these implementations as proof of concept and to fully utilize the speed-up of unlocked updates will require a c++ (libtorch) implementation.

---

### Official Review · Reviewer_ckuE · 2023-10-31

**Soundness:** 3 good
**Presentation:** 2 fair
**Contribution:** 2 fair
**Rating:** 5
**Confidence:** 5

**Summary:**

his paper presents two ideas.
The first is the bidirectional (forward and backward) propagation over a set of synaptic weights.
A part of the network propagates a signal in the forward direction.
The other part propagates the signal in the backward direction.
The second is local blocking or breaking down of a deep network into smaller blocks.
This promotes parallel processing.

**Strengths:**

This paper is well-organized.

**Weaknesses:**

1). The main contributions or claims of this paper seem minimal relative to existing work in this area.

Here are my reasons:

The bidirectional propagation overlaps with the work presented in the paper titled "Bidirectional Backpropagation".
It uses a set of synaptic weights for forward and backward propagation.
This generalizes to the case of using two separate networks.
The two separate network case is a special case that uses a deterministic dropout along the forward and backward propagation.


They also present deep-neural blocking as a method for breaking down deep-neural networks into blocks of smaller networks.
They used the multiplication theorem for probability to factor the complete likelihood to a product of block likelihoods.
You can find this in their paper on "Bidirectional backpropagation for high-capacity blocking networks".
Equations (1) and (2) of their paper show the likelihood factorization.
This is similar to what you have listed as one of the main contributions.
You can also check equation (133) in the paper on "Noise can speed backpropagation learning and deep bidirectional pretraining".

Finally, their paper on "Bidirectional backpropagation for high-capacity blocking networks" combines bidirectional backpropagation with blocking.
This is a combination of bidirectional propagation and deep-neural blocking.

**Questions:**

Please can you identify the distinction(s) between your work and the prior work listed above on bidirectional backpropagation and neural blocking?

---

> ### Author Response · Authors · 2023-11-21
>
> The main contribution of our model is to show that using separated forward and backward networks, in a variational learning setup, local learning rules can be derived (Equations 6 and 7), that do not suffer from the locking and weight transport problem.  Updates are calculated and applied on a per-block basis after a forward pass through the network. The backward network can have any architecture including sharing parameters with the forward network, however we didn't consider that case yet. Hence we would consider this to be the most general case. Also Bidirectional backpropagation doesn't interpret the outputs of the network as exp. family distribution parameters. Furthermore, our method doesn't include a network-wide finetuning phase as in Bidirectional backpropagation. The goal of Bidirectional backpropagation also doesn't seem to be solving the locking and weight transport problem. We will be happy to point out these differences in more details in a revised version of the paper.

---

### Official Review · Reviewer_g9PU · 2023-10-31

**Soundness:** 3 good
**Presentation:** 2 fair
**Contribution:** 3 good
**Rating:** 6
**Confidence:** 3

**Summary:**

The paper introduces a probabilistic framework for block/layer-wise learning, such that a network encodes (parameters of) conditional distributions between latent variables that sequentially go from input to output.

**Strengths:**

1. As far as I know, the idea is novel and is reasonably backed by the variational methods literature.
2. The idea works on simple tasks, and doesn't fail (although doesn't excel) on ImageNet, which is a good sign.

**Weaknesses:**

1. I found the paper rather hard to read. I think the main issue is that the network architecture and its specific computations are unusual, but they're never presented in one place. It takes from page 4 to page 7 to introduce the full architecture/losses. This is fine, since there are many non-trivial steps, but having them in one place would help a lot (perhaps in an extended Tab. 1, which by the way should be labelled as an algorithm).

2. The following claim from the abstract: "demonstrating state-of-the-art performance using block-local learning" is not correct -- ~54% top1 accuracy on ImageNet from Tab. 2 with a ResNet50 is not very good. It's nice to see that the model doesn't fail, but it's not SoTA at all (even AlexNet reaches 56% top1 accuracy).

3. It's also worth noting that block/layer-wise learning can perform as well as backprop, and with a similar idea of using small backward error networks. See Fig. 1 and overall results in [Belilovsky et al., 2019] (cited in this paper). It's not clear if the performance difference is due to inherent problems with the probabilistic interpretation in this paper or some other reasons.

**Questions:**

**Comments**:
1. First of all, the authors should use the ICLR style (currently the typeface is wrong and citations are not highlighted) and fit the paper into 9 pages.
2. Top-3 for MNIST/CIFAR10 is not a standard metric. Moreover, given 10 classes and the simplicity of the datasets, anything but top-1 is mostly meaningless.
3. In S1.3, I think the first expectation should be just an integral over $z_k$. Expectation adds $p(z_k)$ which shouldn’t be there.
4. Citation issue: Jimenez Rezende et al. (2016) citation accidentally includes the author’s middle name.
5. Eqs. S12-13 and later: should there be brackets starting after the first sum?

**Clarification questions**

2. Eq. 7 should have $\beta_k$ instead of $q_k$, right?

3. This bit on page 6 is confusing:
> furthermore, the loss is local with respect to learning, i.e. it doesn’t require global signals to be communicated to each block. In this sense, our approach differs from previous contrastive methods that need to distinguish between positive and negative samples. In our approach, any sample that passes through a block can be used directly for weight updating and is treated in the same way

The paper was about supervised learning up until now, right?

**Conceptual questions**:

1. What’s the actual speed-up compared to backprop? With a naive implementation that doesn't account for unlocked updates, it should be as expensive. I guess it might be possible to put different blocks on different devices so the backward pass in earlier blocks will happen early on, but that wouldn’t work on a single GPU (right?) and also wouldn’t be easy to implement in PyTorch. How was it implemented by the authors?

2. Judging by Fig. 2, posterior bootstrapping introduces backward locking. Is that correct?




-----------
**Post-rebuttal**: since my questions were addressed, but the weaknesses like performance are still there, I'm increasing the score from 5 to 6. (Note to authors: sorry, I accidentally didn't change the rating when changing the review at the end of the discussion period. Corrected now.)

---

> ### Author Response · Authors · 2023-11-21
>
> *Comments*
> Thank you for pointing out these style issues. We have corrected them and revise the pdf.
>
> *Clarification questions*
> 2. Actually, it should be $\rho_k$ as in Eq. (4). Technically $q_k$ is correct here, however, the definition of $q_k$ comes only late in the supplement. We noticed an ambiguity in our use of the distribution $q$. We will simplify the notation to resolve that issue.
>
> 3. Correct. However, a number of previous methods used a contrastive step to arrive at block-local learning rules. Here, no such contrastive step is needed. We will clarify this.
>
> *Conceptual questions*
> 1. As mentioned in the response to ACie, we implemented 2 versions of our algorithm, one a naive implementation which is as expensive as backprop (with additional linear layers for the backward network) and second , as the reviewer rightly suggested to put different blocks on multiple devices. However the updates are still locked due to pytorch’s limitations. We consider these implementations as proof of concept and to fully utilize the speed-up of unlocked updates will require a c++ (libtorch) implementation.
>
> 2. No there is no backward locking in our method. All updates can be computed in forward mode and can be parallelized. The only required synchronization is in the forward path through the network, which is unavoidable. We explained the precise structure of the parallelism in Fig.3.

---

> > ### Comment · Reviewer_g9PU · 2023-11-21
> >
> > Thank you for the response and clarifications! I have further questions though.
> >
> > First, I think it might be a good idea to updated the pdf during the discussion period to fix the style issues.
> >
> > Second, I'm still confused about backward locking and bootstrapping. Fig. 2, as I understand it, should the optimal bootstrapping schedule that propagates errors backwards, which naturally introduces backwards locking (right?). Fig. 3 shows the simplified bootstrapping schedule that doesn't have that issue, but as mentioned in the appendix is only an approximation. From what I understood the experiments are done for both the simplified and the optimal schedules (e.g. Tab. S1 mentions the optimal one). Is this correct?

---

> > > ### Author Response · Authors · 2023-11-22
> > >
> > > 1) Sure, we uploaded the new pdf fixing the style issue.
> > >
> > > 2) There still seems some confusion. $\rho$ are the variational posteriors which are computed based on forward and backward messages $\alpha$ and $\beta$ (Eq. 4). Since $\beta$ can be computed directly based on the targets $y$ there is no locking introduced (Fig. 1). Therefore, all bootstrapping methods presented parallelize according to Fig.3, there is no locking as long as the structure in Fig. 1 is followed.

---

> > > > ### Comment · Reviewer_g9PU · 2023-11-22
> > > >
> > > > Thank you! Yes, that clarifies the confusion. I've increased the score from 5 to 6.

---

### Official Review · Reviewer_nrns · 2023-11-03

**Soundness:** 3 good
**Presentation:** 2 fair
**Contribution:** 3 good
**Rating:** 6
**Confidence:** 3

**Summary:**

This paper introduces a probabilistic interpretation of the input-output mapping defined by a neural network, together with a variational inference method used to optimize the log-likelihood.
The variational loss is an upper bound on the actual log-likelihood, and thus optimizing this variational loss to zero guarantees the optimization of the log-likelihood.
The variational loss can be decomposed into block local loss terms, making the optimization of this surrogate objective amenable to parallelization, removing both forward and backward locking of the back-propagation algorithm.
It also solves the weight transport problem as the feedback from labels, usually computed with back-propagation, is provided by a separate neural network.
The resulting algorithm shows promising performances on standard classification benchmarks, with accuracy being rather low with respect to vanilla gradient descent implemented with the back-propagation algorithm.
The proposed method being however well-posed and grounded in variational inference, where practical issues are well documented, it is likely that further improvements could be achieved by optimizing all the hyperparameters such as the design of the feedback network, the expressivity of the exponential family considered as an intermediate representation, or simply the optimization hyperparameters such as the learning rate or the type of optimizer used.

The paper allocates a long portion of the main body to the description of the proposed method and only shows a brief summary of the experiments performed, with experimental details and some additional experiments (i.e. ablation study) deferred in the appendix.

**Strengths:**

1) The probabilistic interpretation of input-output mapping allows the derivation of a variational inference method to tackle the optimization of the model weights. The reviewer is not aware of such formulation in the literature but is however not an expert in this particular field.
2) The proposed method is broadly applicable to most feedforward architectures used in deep learning applications.
3) The results on classification benchmarks are on par with other local learning methods without extensive hyperparameter tuning.

**Weaknesses:**

1) The proposed probabilistic interpretation of the input-output mapping appears new and is a bit hard to follow at first glance.
2) It is difficult to understand how the variational distribution $q$ is actually defined. Only backward messages $\beta_k = q(y | z_k)$ and it is unclear how the full posterior $q(z_k | x, y)$ is defined. My best guess is that it is implicitly defined through the bayes rule by combining the messages $\beta_k$ for all $1 \leq k \leq N$.
3) Some notations are only implicitly defined, such as $q_k$, or not defined at all such as $a_{kj}$ and $z_{kj}$.
4) It is difficult to understand in which case the variational posterior could recover the true posterior. In other words, it is unclear which messages $\beta_k$ would lead to a perfect reconstruction of the true posterior, as well as how to compute them.
5) There are no guidelines on how to design the different components of the proposed algorithm. For example, there is no ablation study on the expressivity of the EF distribution used as a latent representation, nor on the expressivity of the feedback network.
6) [Minor comment] Equation S2 of Section S1.2 in the supplement material defines an upper bound of the log-likelihood and refers to equation (1) of the main text. However, equation (1) of the main text refers to the gradient of the log-likelihood.

**Questions:**

1) The distribution $q$ is only defined implicitly through the definition of backward messages $\beta_k = q(y | z_k)$. Could the author either confirm that it is only implicitly defined because we only need to compute messages $\beta_k = q(y | z_k)$ or specify the definition of the full posterior $q(z_k| x, y)$?
2) In equation (5), I don’t understand what the index $j$ stands for. Are $\alpha_{kj}$ and $\beta_{kj}$ parameters of the distribution $\alpha_k$ and $\beta_k$ or actual distributions in their own right?
3) Does equation (6) make use of the assumption that $p_k$ and $\alpha_k$ are Gaussian or from a distribution in the exponential family?
4) Could the author clarify the approximation gap between the true log-likelihood and their variational surrogate? In particular, when does the variational loss recover the true log-likelihood?
5) Could the authors give some intuition on the design of the latent representation and the expressivity needed in the feedback network?

---

> ### Author Response · Authors · 2023-11-21
>
> 1. We tried to keep the initial part of the derivation general, therefore $\beta$ is initially only defined as any distribution $q(y|z_k)$. We use a variational posterior that factorizes into the parts $\alpha$ and $\beta$ according to Eq.(4). $\alpha$ is the true forward message computed by the forward network. $\beta$ is the approximate backward message that determines how well the network activation at block $k$ predicts the labels $y$.
> Based on this general derivation, to arrive at a concrete implemenation we used the exponential family assumption. The variational posterior that was used in our experiemnts is defined in Eq. S19.
>
> 2. $\alpha_{kj}$, $\beta_{kj}$ are the independent components corresponding to activations of individual units of the forward and backward network, respectively. They determine the natural parameters  of distributions $\alpha_k$, $\beta_k$.
>
> 3. Eq. 6 is the general form for any member of the exp. family. Conveniently, only the first and second order statistics are needed to compute the divergence and $\mu$ and $\sigma$ refer here to these sufficient statistics. We establish this result in section S1.4 in detail. In experiments we used a Gaussian distribution, for which the derivation is in S1.5.
>
> 4. The approximation becomes exact when if $\beta_k = p(y|z_k)$, i.e. the true posterior of the targets $y$, given the network output at block $k$, according to the forward network (its true Bayesian inversion). However, with the simplified form of variational distribution that we use, where $\beta_k$’s are given by backward network activity, a gap between the true log-likelihood  and the variational surrogate will always remain, as is the case in standard variational methods.
>
> 5. We chose the simplest possible feedback network that did not need additional gradient based updates for efficiency. Early tests showed that more sophisticated architectures were not needed in those cases but it is still a future work. However, more complex tasks like ImageNet may require more complex backward network structures, which may explain the gap in performance.

---

> > ### Comment · Reviewer_nrns · 2023-11-22
> >
> > I thank the author for clarifying some of my concerns.
> > In general, I still have concerns on how to efficiently tune the proposed method to optimize model performance, however, the presented results are encouraging given that the formulation of the learning problem is local and that the variational upper bound allows to implicitly optimize the true classification loss, contrary to other local learning approach.
> > I will raise my score to 6 since I think this paper should be accepted because, to my knowledge, such variational framework for local learning is novel in the literature and the interpretation of the activation of the different modules are also novel.
> > Thus, I consider that the provided amount of work is enough for further research to be conducted by peers.
> > I am not increasing my score further as there are not enough ablation on the flexibility of the proposed method, particularly on trainable feedback network with higher statistical capacity.
> > It should be noted that DGL method achieved higher test accuracies on ImageNet despite optimizing a greedy objective at each block [1].
> >
> > [1] Belilovsky, Eugene, Michael Eickenberg, and Edouard Oyallon. "Decoupled greedy learning of cnns." International Conference on Machine Learning. PMLR, 2020.

---

### Author Response · Authors · 2023-11-21

We thank the reviewer for their comments. We will update the draft to explain the method more clearly. Please find the answers to the questions below.

---

### Author Response · Authors · 2023-11-22
**Reproducibility: Model and training scripts**

Pytorch code for training Imagenet is available at anonymous repository: https://github.com/CeFKZP/congenial-octo-tribble .
Same model and loss was used to train Cifar10 as well.

---

### Meta-Review · Area_Chair_LcU4 · 2023-12-05

**Metareview:**

This paper proposes a block-local learning algorithm via a feedback network that decouples forward and backward weights. The reviewers found strength in aspects of the paper, such as its broad applicability and the promising results on simple settings. The discussion highlighted connections to some prior work, which, while partially addressed by the authors, diminishes some of the novelty of this work. Several reviewers found the presentation to be unclear and the method difficult to follow, owing to the organization, notation, and explanation of ideas. The organization of the paper could also be improved, with a more concise and interpretable description of the method leaving more room for further experiments and analysis.

**Justification For Why Not Higher Score:**

Insufficient clarity in presenting the main methods and lack of sufficiently compelling experimental results.

**Justification For Why Not Lower Score:**

N/A

---

### Decision · Program_Chairs · 2024-01-16

Reject